# Characterization of Intervertebral Disc Changes in Asymptomatic Individuals with Distinct Physical Activity Histories Using Three Different Quantitative MRI Techniques

**DOI:** 10.3390/jcm9061841

**Published:** 2020-06-12

**Authors:** Daniel L. Belavy, Helena Brisby, Benjamin Douglas, Hanna Hebelka, Matthew J. Quittner, Patrick J. Owen, Timo Rantalainen, Guy Trudel, Kerstin M. Lagerstrand

**Affiliations:** 1School of Exercise and Nutrition Sciences, Institute for Physical Activity and Nutrition, Deakin University, Geelong, 221 Burwood Highway, Burwood VIC 3125, Australia; d.belavy@deakin.edu.au (D.L.B.); matt.quittner@deakin.edu.au (M.J.Q.); p.owen@deakin.edu.au (P.J.O.); 2Sahlgrenska Academy, Institute of Clinical Sciences, University of Gothenburg, PO Box 426, SE405 30 Gothenburg, Sweden; helena.brisby@vgregion.se (H.B.); hanna.hebelka@vgregion.se (H.H.); 3School of Exercise and Nutrition Sciences, Deakin University, Geelong, 221 Burwood Highway, Burwood VIC 3125, Australia; bendouglas1927@gmail.com; 4Faculty of Sport and Health Sciences, University of Jyväskylä and Gerontology Research Center, PL 35, 40014 Jyväskylä, Finland; timo.rantalainen@jyu.fi; 5Bone and Joint Research Laboratory, Department of Medicine, Division of Physical Medicine and Rehabilitation, University of Ottawa, 505 Smyth Rd, Ottawa, ON K1H 8M2, Canada; gtrudel@toh.ca

**Keywords:** intervertebral disc, magnetic resonance imaging, sport medicine, T2-mapping, Dixon

## Abstract

(1) Background: Assessments of intervertebral disc (IVD) changes, and IVD tissue adaptations due to physical activity, for example, remains challenging. Newer magnetic resonance imaging techniques can quantify detailed features of the IVD, where T2-mapping and T2-weighted (T2w) and Dixon imaging are potential candidates. Yet, their relative utility has not been examined. The performances of these techniques were investigated to characterize IVD differences in asymptomatic individuals with distinct physical activity histories. (2) Methods: In total, 101 participants (54 women) aged 25–35 years with distinct physical activity histories but without histories of spinal disease were included. T11/12 to L5/S1 IVDs were examined with sagittal T2-mapping, T2w and Dixon imaging. (3) Results: T2-mapping differentiated Pfirrmann grade-1 from all other grades (*p* < 0.001). Most importantly, T2-mapping was able to characterize IVD differences in individuals with different training histories (*p* < 0.005). Dixon displayed weak correlations with the Pfirrmann scale, but presented significantly higher water content in the IVDs of the long-distance runners (*p* < 0.005). (4) Conclusions: Findings suggested that T2-mapping best reflects IVD differences in asymptomatic individuals with distinct physical activity histories changes. Dixon characterized new aspects of IVD, probably associated with IVD hypertrophy. This complementary information may help us to better understand the biological function of the disc.

## 1. Introduction

The intervertebral disc (IVD) consists of the outer annulus fibrosus and the gelatinous central nucleus pulposus [1]. The high content of type 1 collagen-containing fibers in the annulus fibrosus creates a strong fibrous ring, whereas the nucleus pulposus matrix is built up of type 2 collagen and water-binding proteoglycan molecules. The IVD degenerates with aging and if injured, structural changes might precipitate and maintain lower back pain [2]. The degeneration of discs within the lumbar spine starts early in life, as early as 20 years [3]. Thus, to measure subclinical changes within the IVD tissue might be of value to gain increased knowledge regarding the complex degenerative process and to understand the more advanced stages of degeneration. Furthermore, imaging data that can be linked to degeneration and used as markers for long-term follow-up could have wide applications in the search for novel therapies of the disc.

Qualitative or categorical scales have been used to grade IVD degeneration (e.g., the Pfirrmann scoring system [4]). In addition, quantitative MRI techniques have been developed that can detect degenerative IVD changes on a continuous scale of measurement. [5]. Additionally, advanced analysis software has enabled the automated quantification of the IVD degeneration, as well as the detailed characterization of different degeneration patterns [6]. However, comparisons of the usefulness of quantitative MRI have received little attention.

T2-mapping is commonly used in spine research for the characterization of degenerative disc changes. With T2-mapping, objective and quantitative measures of the disc are provided, reflecting both water content and the orientation of collagen fibers. Moreover, the group of Belavy has recently reported significant differences in IVD composition between groups with different training histories [7,8,9] and has, from a randomized controlled trial in patients with lower back pain, evaluated the effect of exercise recommended alternative MRI markers of IVD changes [10].

Another quantitative MRI technique for the characterization of degenerative disc changes is T2-weighted (T2w) imaging [11,12,13]. Similar to T2-mapping, T2w imaging relies on T2 relaxation and, as such, reflects the water content as well as the matrix structure of the IVD [14]. T2w imaging is not currently used in the clinic to quantify IVD changes, since the contrast is influenced by several factors, including machine type, sensitivity of the coil, and subject position, which limit its usefulness. In the clinical setting, however, T2w imaging is routinely used for the visualization of the spine and has the advantage of being robust as well as fast in comparison with T2-mapping. As such, T2w imaging is an attractive technique to further adapt as a tool for the quantification of degenerative disc changes. The Dixon technique [15], not previously applied to IVD evaluation, has been extensively used in the literature to quantify the fat content of the liver. Technological advances have recently been incorporated into the technique [16], providing resources for the robust reconstruction of water/vascular regression images. As such, Dixon imaging should be able to detect differences in the water content of the IVD without reflecting the structure of the IVD matrix and, thus, reflect more aspects of disc degeneration than T2-mapping and T2w imaging techniques. Both this and the very short scan time are promising features for clinical use in the IVD evaluation.

In this study, we investigated the performances of the three different quantitative techniques, T2-mapping and T2w and Dixon imaging, to characterize IVD differences in asymptomatic individuals with distinct physical activity histories.

## 2. Material and Methods

### 2.1. Ethical Approval and Subjects

This study was a secondary analysis of an existing dataset [9,10] on the impact of physical activity and inactivity on the spine in an asymptomatic population. The data set consisted of 101 participants, 54% women and 46% men, mean (SD) age 30.0 (3.6) years, 173.5 (9.6) cm height, 69.9 (13.4) kg mass, divided into four groups of individuals with distinct physical activity histories, but similar Pfirrmann grade distributions. Only people with a minimum of five years of history at their current physical activity level were included in the study: either no sport (sedentary referents), cyclists reporting a minimum of 150 km cycled per week (high-volume road cyclists), runners reporting 20–40 km running per week (joggers), or runners reporting 50+ km running per week (long-distance runners). See previous work [9,10] for further details.

Exclusion criteria included current spinal (cervical, thoracic or lumbar) pain, history of spinal surgery, history of traumatic injury to the spine, known scoliosis for which prior medical consultation had been sought, current or prior smoker, known claustrophobia and possible pregnancy.

All subjects gave their informed consent for inclusion before they participated in the study. The study was conducted in accordance with the Declaration of Helsinki. The study was approved by the Deakin University Faculty of Health human ethics advisory group.

### 2.2. Testing and MR Scanning Protocol

Participants were instructed to perform no exercise on the day of their scan. To avoid the impact of normal diurnal variation on the spine [17], all testing was performed after midday. Upon arriving at the radiology facility, participants were required to sit for a minimum of 20 min prior to entering the scanner to help standardize the IVD hydration status. Participants sat for a mean (SD) of 46 (17) min. Spine volume from the T11 vertebral body to the sacrum was imaged in all subjects using a 3T Phillips Ingenia scanner (Amsterdam, The Netherlands). Three different scan protocols were used:Sagittal T2-mapping using spin-echo multi-echo sequences (8 echo times: 15.75, 36.75, 57.75, 78.75, 99.75, 120.75, 141.75 and 162.75 ms; repetition time: 2000 ms; number of slices: 12; slice thickness: 3 mm; interslice distance: 1.5 mm; field-of-view: 281 × 281 mm; resolution: 0.366 mm per pixel; acquisition time: 9 min 30 s).Sagittal T2w imaging (echo time: 70 ms; repetition time: 2600 ms; number of slices: 15; slice thickness: 3 mm; interslice distance: 1.5 mm; field-of-view: 357 × 357 mm; resolution: 0.532 mm per pixel; acquisition time: 3 min).Sagittal Dixon imaging (echo times: 2.45/3.67 ms; repetition time: 5.27 ms; number of slices: 20; slice thickness: 3 mm; interslice distance: 0 mm; bandwidth: 500; field-of-view: 400 × 400 mm; resolution: 0.833 mm per pixel; acquisition time: 1 min 20 s). A Dixon technique with asymmetrical echoes (mDIXON) was utilized [16].

### 2.3. Image Analysis

To ensure the blinding of the examiner in offline image measurements, each subject was assigned a random numeric code (obtained from www.random.org). A radiologist determined the Pfirrmann grade of each lumbar IVD using the T2w images. All Pfirrmann grades, except Pfirrmann grade 5 IVDs, were observed in the cohort. An off-line software was used to perform all quantitative MR measurements and a custom written ImageJ plugin was used to implement the IVD measurements after manual segmentation of the IVD using the native “polygon selections” tool in ImageJ.

IVDs T11/12 to L5/S1 were included in the analysis. Supernumerary lumbar IVDs (L6/S1; eight individuals) were excluded. One hypoplastic L5/S1 IVD at a sacralized L5 vertebral level was included in analysis. The image number corresponding to each vertebral and IVD level was noted. IVDs were segmented in five subregions from the anterior to posterior aspect of the disc, as shown in Figure 1 [15,16,18]. Then, a custom written plugin measured the signal intensity of the whole IVD as well as the five subregions.

Depending on the MRI technique, the following calculations were performed:

T2-mapping: To reconstruct T2-maps, the T2-time in each pixel was calculated from the spin-echo multi-echo images using a linear fit to the natural logarithm of the image intensity in each of the eight MR echoes.T2w imaging: For normalization, the ratio of the average T2w signal intensity (T2w-SI) in the nucleus to the average T2w-SI in the anterior and posterior subregions (annulus) was calculated.Dixon imaging: An iterative algorithm with least-squares estimation was used to maximize the noise performance of the water signal [16].

A detailed analysis of all MR images was then performed. In addition to the mean values of the whole IVD, values of the nucleus sub-region and values of the ratio of the nuclear subregion to anterior and to posterior subregions (ratio nucleus/annulus) were determined for all techniques using the central three sagittal slices to capture the central (most hydrated) region of the nucleus to be comparable to prior works [11,12].

Moreover, three-dimensional (3D) plots of the IVD were generated for all MRI techniques. To generate the plots, the pre-processed data for the T2-mapping, T2w imaging and Dixon imaging were interpolated across the width of the IVD in each subregion.

### 2.4. Statistical Analyses

To examine the sensitivity of each MRI technique to differentiate between IVD characteristics in people with distinct training histories, IVD variables in each athletic group were compared to the sedentary referent group using unpaired *t*-tests. Sub-analysis was also performed to examine the covariation of the MRI techniques, and Pearson’s product moment correlation between each variable was calculated. Additionally, unpaired *t*-tests were used to examine the difference in variables obtained from each MRI technique on Pfirrmann grades 2, 3 and 4 IVDs compared to the Pfirrmann grade 1 IVDs.

An alpha level of 0.05 was considered statistically significant. The “R” statistical environment (version 2.10.1) was used for all analyses.

## 3. Results

### 3.1. The T2-Mapping and T2w Imaging Techniques

As previously reported [9,10], the T2-time by the T2-mapping of the whole IVD and nucleus, as well as the nucleus/annulus ratio, distinguished between the various participants’ training histories, as shown in Table 1 and Figure 2. These subtle differences between groups were not found with T2w imaging. The T2-mapping and T2w imaging techniques displayed large regional differences over the IVDs, with higher values centered at the nucleus, as shown in Figure 3. The T2-time and T2w-SI of the whole IVD and nucleus were decreased for Pfirrmann grades 2, 3 and 4 in comparison with grade 1, as shown in Table 2 and Figure 4. The nucleus/annulus ratio was also decreased in grades 2, 3 and 4 for the T2-time, but only in the higher Pfirrmann grades, 3 and 4 for T2w-SI, as shown in Table 2 and Figure 4.

### 3.2. The Dixon Imaging Technique

In comparison with the sedentary group, higher water content by Dixon imaging of the IVD was detected in the long-distance runners, but not in groups with other training histories, as shown in Table 1 and Figure 2. The Dixon imaging technique displayed small and non-significant differences in the water content over the IVD and between nucleus and annulus, as shown in Figure 3. The nucleus/annulus ratio was close to 1, as shown in Table 2. The water content was also shown to be insensitive to general degeneration changes. The whole IVD and the nucleus water content only detected differences between Pfirrmann grades 4 and 1, as shown in Table 2 and Figure 4.

### 3.3. Correlation Analyses

There was a strong correlation between T2-mapping and T2w imaging regarding whole IVD and a moderate correlation (*r* = 0.66, *p* < 0.001) regarding nucleus/annulus ratio. For Dixon imaging, the whole IVD and the nucleus correlated only weakly (*r* all < 0.29) with the T2-time and the T2w signal, as shown in Table 3.

## 4. Discussion

In this study, three quantitative MRI techniques (T2-mapping, T2w imaging and Dixon imaging) were used to characterize the IVDs in a young, asymptomatic, population with different training histories. Compared with T2w imaging, T2-mapping was the most sensitive technique to detect IVD changes. While there was a good correlation between the T2-mapping and the T2w imaging technique, only T2-mapping detected differences associated with physical activity history in cyclists, joggers and long-distance runners, compared to the sedentary referents. Additionally, only T2-mapping consistently distinguished between Pfirrmann grades 1 and 2.

To our knowledge, this is the first work that evaluated the feasibility of the Dixon imaging technique to characterize IVD hydration. As shown in this study, long-distance runners presented significantly higher water content than the other groups. All groups displayed differences in the T2-value, but only in the long-distance runners did the IVD changes correspond to an increased water signal. The Dixon images did not display large differences in the water content between nucleus pulposus and annulus fibrosus. In fact, the images displayed only a slow variation across the IVD, contrary to the T2-maps and T2w images that display large drops at the outer edges of annulus fibrosus where the water is strongly bound to the extracellular matrix. This can be explained by Dixon imaging only detecting the water content without reflecting the structure of the IVD matrix as both T2 contrast techniques do. Moreover, Dixon imaging did not correlate with the Pfirrmann scoring system or with the other MRI techniques. Hence, the technique characterized aspects of the IVD structure which complemented the information obtained by the other techniques. Previous studies on the impact of physical activity and inactivity on the spine in an asymptomatic population report significant differences in IVD height relative to the vertebral bodies in long-distance runners [9,10]. The increased height might be an indicator of IVD hypertrophy, measured as an increased water content in this cohort. Future studies are encouraged to validate this finding and investigate the additional value of including MRI techniques for water content quantification in spine examination protocols.

IVD images generated with T2-mapping display the water content of the IVD [18]. The T2-time obtained by T2-mapping has also been shown to depict the structural integrity of the disc matrix [19], displaying a stronger correlation with collagen structure than with changes in the cartilage matrix of the IVD. Hence, both the whole IVD and the nucleus T2-values should be able to reflect changes due to IVD degeneration. Our findings confirmed this and showed a high correlation between IVD T2-times and degeneration, according to the Pfirrmann scoring system. The significantly higher T2-times measured in physical trained individuals might indicate reduced IVD degeneration, as previously reported [9,10]. The T2w signal intensity also reflects the IVD T2-time, but only qualitatively. However, by normalizing the nucleus T2w signal with the annulus signal, the IVD evaluation was improved but with limited sensitivity compared with the T2-mapping technique. The limited sensitivity may stem from the heterogeneity of the coil sensitivity. Our findings suggest that T2-mapping should be the preferred T2 contrast technique to characterize preclinical degenerative IVD changes in asymptomatic individuals. This will permit research advances not only in determining risk factors for IVD degeneration with age or with various lifestyle and pathological situations but will also allow intervention trials. Challenges with the T2-mapping technique prevent its implementation into clinical practice. T2-mapping is slower than current clinical T2w imaging techniques, taking approximately 10 min per scan, and the complex postprocessing of the data is needed to model the MR signal into T2-time. However, advances in artificial intelligence may soon provide the clinician with tools for time efficient and automated T2-mapping analyses.

## 5. Limitations

In this study, the whole IVDs were delineated in addition to five sub-regions, the central nucleus and the anterior and posterior annulus. Current MRI technology cannot produce images that can precisely separate nucleus from annulus. Therefore, it is possible that nucleus tissue was incorporated in the anterior and posterior subregions, labelled as annulus tissue. Anatomical-based manual segmentation was impractical for a study of over 100 patients using three MRI techniques measuring six IVD levels (909 disks). Therefore, multiple sagittal slices were segmented in five subregions [20]. Since this limitation applied to all techniques, it should not affect the inter-technique comparisons, the main goal of our study. Only one radiologist determined the Pfirrmann grade of the IVDs based on published data of the excellent interobserver agreement of the Pfirrmann scoring system [21].

## 6. Conclusions

Among the three quantitative techniques, T2-mapping best reflected changes in IVD degeneration, according to the Pfirrmann scoring system. Most importantly, T2-mapping characterized IVD differences in individuals with different training histories. The Dixon imaging technique identified significantly higher water content in the IVDs of long-distance runners compared to the other training groups, probably associated with IVD hypertrophy. Additionally, Dixon displayed a weak correlation with the Pfirrmann scale and the other MRI techniques. Hence, Dixon may present information on new aspects of the IVD, irrespective of matrix changes and add complementary information to the assessment and understanding of the biology of the IVD.

## Figures and Tables

**Figure 1 jcm-09-01841-f001:**
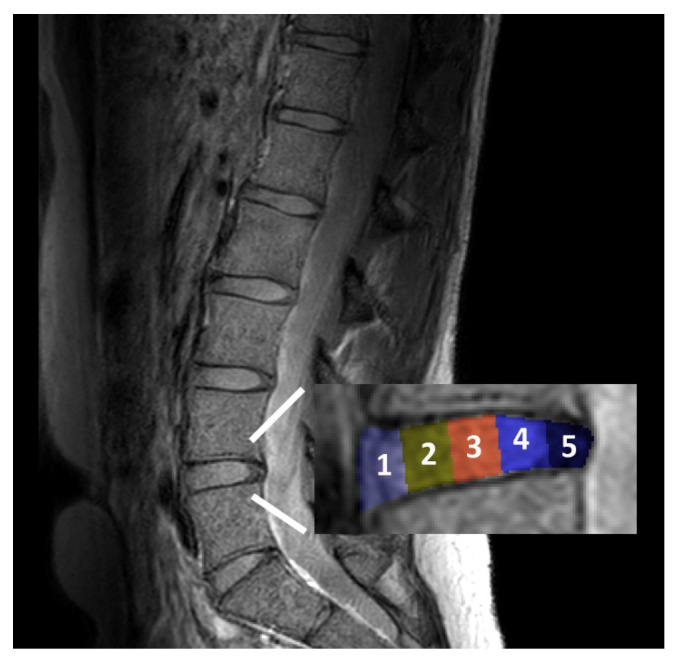
Graphical illustration of the segmentation of the IVD into five subregions from (1) anterior to (5) posterior aspect of the disc.

**Figure 2 jcm-09-01841-f002:**
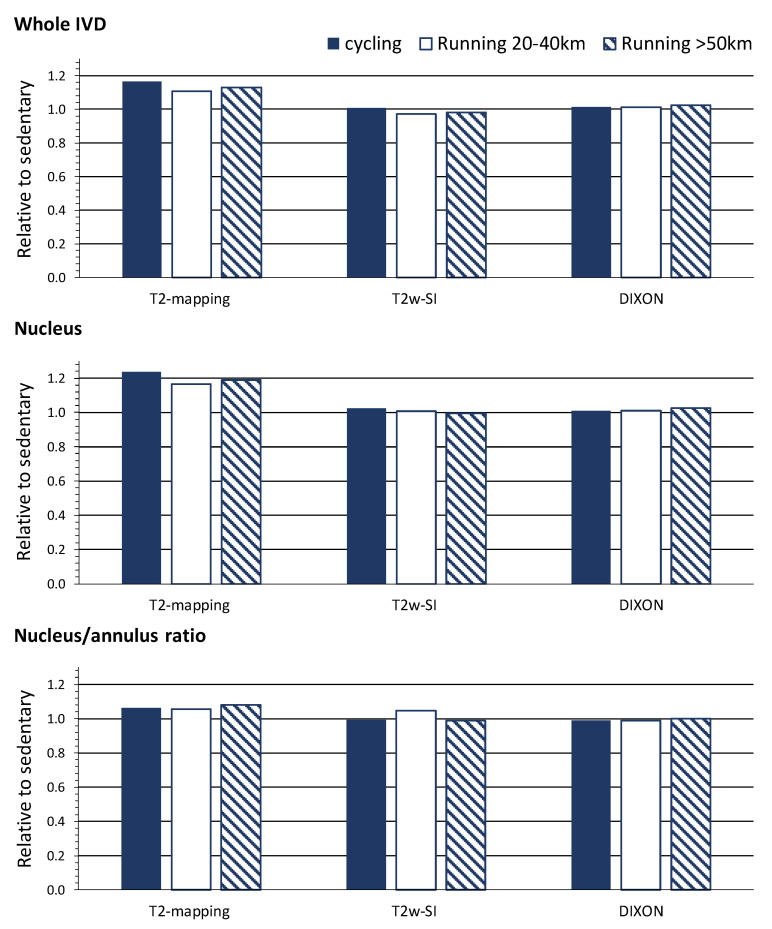
Differences of each MRI technique in quantifying the intervertebral disc. Data are relative differences in physical activity histories compared to the sedentary group. See Table 1 for statistical comparisons.

**Figure 3 jcm-09-01841-f003:**
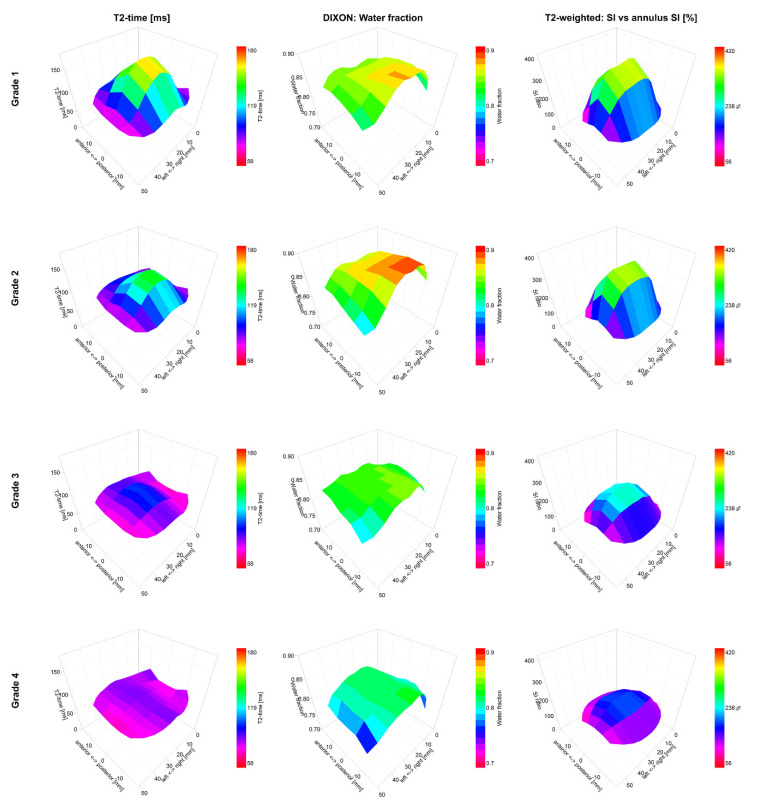
Graphical representations of the IVDs in terms of 3D plots for T2-mapping, T2w imaging and Dixon imaging. Values are averaged across each Pfirrmann grade for all IVDs, T12/L1 to L5/S1 (606 IVDs from 101 subjects). Color keys indicate range of values.

**Figure 4 jcm-09-01841-f004:**
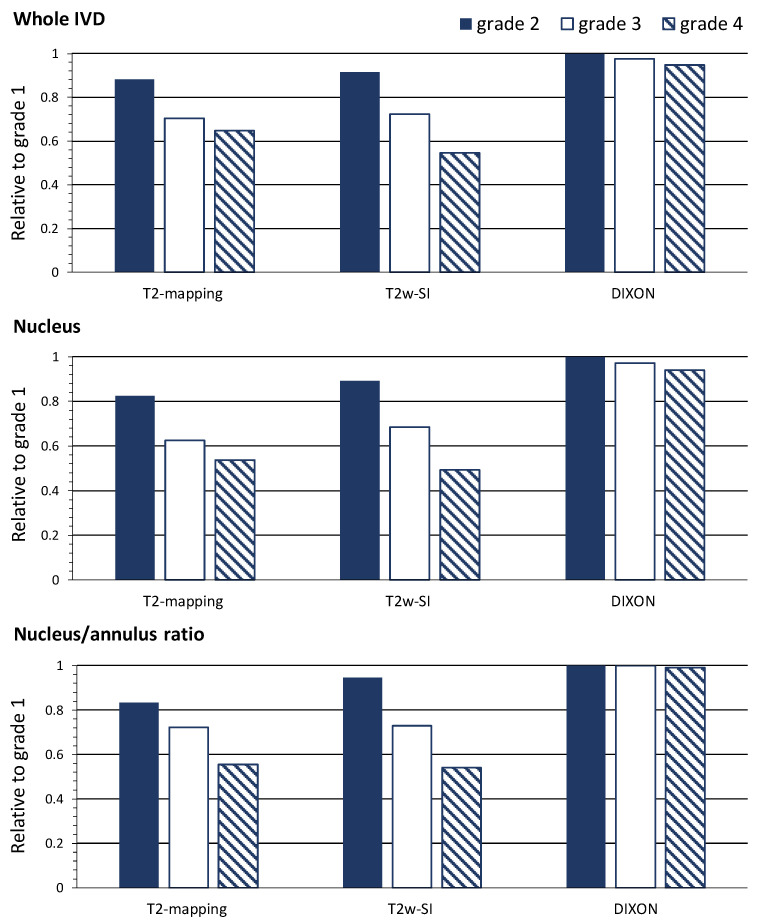
Differences of each MRI technique in quantifying the intervertebral disc. Data are relative differences in Pfirrmann grades 2, 3 and 4 compared to grade 1. See Table 2 for statistical comparisons.

**Table 1 jcm-09-01841-t001:** Differences between T2-mapping, T2w imaging and Dixon imaging according to physical activity histories.

	Participant Group (Number of IVDs)			
	Sedentary (*n* = 144)	High-Volume Cycling (*n* = 132)	Running: 20–40 km (*n* = 180)	Running: >50 km (*n* = 150)
**Whole IVD**				
T2-mapping (ms)	104.4 (1.7)	121.6 (1.7) ^‡^	115.6 (1.5) ^‡^	117.9 (1.6) ^‡^
T2w-SI (no unit)	326.9 (4.9)	329.7 (5.1)	317.6 (4.4)	320.5 (4.8)
Dixon (%)	86.0 (0.6)	87.1 (0.7)	87.1 (0.6)	88.1 (0.6) *
**Nucleus**				
T2-mapping (ms)	111.4 (3.0)	137.7 (3.1) ^‡^	129.8 (2.7) ^‡^	132.6 (2.9) ^‡^
T2w-SI (no unit)	423.6 (8.9)	434.0 (9.3)	426.4 (7.9)	421.6 (8.7)
Dixon (%)	87.2 (0.6)	88.1 (0.7)	88.1 (0.6)	89.4 (0.6) *
**Nucleus/Annulus Ratio**				
T2-mapping (ms)	1.40 (0.03)	1.49 (0.03) *	1.48 (0.02) *	1.51 (0.03) †
T2w-SI (no unit)	3.31 (0.07)	3.29 (0.07)	3.46 (0.06)	3.27 (0.07)
Dixon (%)	1.01 (0.00)	1.00 (0.00)	1.01 (0.00)	1.01 (0.00)

Values are mean (SD). *: *p* < 0.05; ^†^: *p* < 0.01; ^‡^: *p* < 0.001 and indicate significance of difference to the sedentary group. Data from all IVDs, T12/L1 to L5/S1, are presented. IVD: intervertebral disc; T2: relaxation time; T2w-SI: T2-weighted signal intensity.

**Table 2 jcm-09-01841-t002:** T2-mapping and T2w imaging reflected the IVD Pfirrmann grades, but Dixon imaging was less sensitive.

	Pfirrmann Grade (Number of IVDs)			
	1 (*n* = 47)	2 (*n* = 457)	3 (*n* = 60)	4 (*n* = 42)
**Whole IVD**				
T2-mapping (ms)	133.9 (17.3)	118.1 (56.8) ^‡^	94.4 (26.2) ^‡^	86.8 (23.9) ^‡^
T2w-SI (no unit)	368 (42)	337 (138) ^‡^	266 (63) ^‡^	201 (58) ^‡^
Dixon (%)	87.0 (7.3)	87.8 (24.0)	84.8 (11.1)	82.4 (10.1) ^†^
**Nucleus**				
T2-mapping (ms)	166.0 (25.0)	137.1 (81.8) ^‡^	103.9 (37.7) ^‡^	89.0 (34.4) ^‡^
T2w-SI (no unit)	551 (72)	492 (236) ^‡^	377 (109) ^‡^	271 (99) ^‡^
Dixon (%)	87.2 (7.9)	88.1 (25.9)	84.8 (11.9)	81.9 (10.9) ^†^
**Nucleus/Annulus Ratio**				
T2-mapping (ms)	1.8 (0.3)	1.5 (0.9) ^‡^	1.3 (0.4) ^‡^	1.0 (0.4) ^‡^
T2w-SI (no unit)	3.7 (0.7)	3.5 (2.3)	2.7 (1.1) ^‡^	2.0 (1.0) ^‡^
Dixon (%)	1.00 (0.03)	1.01 (0.10)	1.00 (0.05)	0.99 (0.04)

Values are mean (SD). ^†^: *p* < 0.01; ^‡^: *p* < 0.001 and indicate significance of difference in mean to the Pfirrmann grade 1 category. Data from all IVDs, T12/L1 to L5/S1, (606 discs from 101 subjects) are presented. IVD: intervertebral disc; T2: relaxation time; T2w-SI: T2-weighted signal intensity.

**Table 3 jcm-09-01841-t003:** Quantification of the IVD by T2-mapping correlated moderately with T2w-SI and weakly with Dixon.

		Whole IVD			Nucleus			Nucleus/Annulus Ratio		
		T2-mapping	Dixon	T2w-SI	T2-mapping	Dixon	T2w-SI	T2-mapping	Dixon	T2w-SI
**Whole IVD**	T2-mapping		0.10 *	0.5 ^‡^	0.93 ^‡^	0.10 *	0.46 ^‡^	0.43 ^‡^	0.05	0.24 ^‡^
	Dixon			0.2 ^‡^	0.13 ^†^	0.99 ^‡^	0.25 ^‡^	0.16 ^‡^	0.38 ^‡^	0.16 ^‡^
	T2w-SI				0.56 ^‡^	0.29 ^‡^	0.92 ^‡^	0.56 ^‡^	0.16 ^‡^	0.49 ^‡^
**Central IVD**	T2-mapping					0.13 ^†^	0.61 ^‡^	0.69 ^‡^	0.03	0.41 ^‡^
	Dixon						0.26 ^‡^	0.16 ^‡^	0.51 ^‡^	0.16 ^‡^
	T2w-SI							0.73 ^‡^	0.13 ^†^	0.74 ^‡^
**Nucleus/** **Annulus Ratio**	T2-mapping								0.03	0.66 ^‡^
	Dixon									0.05
	T2w-SI									

Values are Pearson’s correlation co-efficient. Data from all IVDs, T12/L1 to L5/S1, (606 discs from 101 subjects) are presented. Significance of the correlation was indicated as *: *p* < 0.05; ^†^: *p* < 0.01; ^‡^: *p* < 0.001. IVD: intervertebral disc; T2: relaxation time; T2w-SI: T2-weighted signal intensity.

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
