# Peer review of "Characterization of Intervertebral Disc Changes in Asymptomatic Individuals with Distinct Physical Activity Histories Using Three Different Quantitative MRI Techniques"

_jcm, 2020, doi:10.3390/jcm9061841_

Round 1

Reviewer 1 Report

Figure 1: While I like the use the 3D plots, I think the incorporation of the sagittal view and the 5 segments (as seen in the Scientific Reports) will help aid readers.

Author Response

Please see your comments and our reply below, as well as changes in the revised manuscript (attached file), indicated in grey.

Figure 1: While I like the use the 3D plots, I think the incorporation of the sagittal view and the 5 segments (as seen in the Scientific Reports) will help aid readers.

Thanks for your suggestion, we have included a figure of the sagittal view and the 5 segments in Material and Methods section to describe the segmentations of the IVDs for reconstruction of the 3D plots. In the revised manuscript, the new included figure is denoted Figure 1, changing the 3D plot from Figure 1 to Figure 2.

Reviewer 2 Report

The authors compare three types of different quantitative MRI techniques to assess intervertebral disc changes in asymptomatic adults, with distinct physical activity histories.

The work is very interesting, especially since it refers to asymptomatic people, of different kinds of activity, and shows distinct possibilities to assess changes in the IVD.

It would be very important to develop specific standards in order to assess and predict early degenerative changes in the intervertebral disc.

Author Response

Please see your comment and our reply below.

The authors compare three types of different quantitative MRI techniques to assess intervertebral disc. The work is very interesting, especially since it refers to asymptomatic people, of different kinds of activity, and shows distinct possibilities to assess changes in the IVD. It would be very important to develop specific standards in order to assess and predict early degenerative changes in the intervertebral disc

Yes, we agree that it is very important to develop specific standards to assess and predict early degenerative changes in the intervertebral disc. We hope our findings can contribute knowledge to such work.

Reviewer 3 Report

The paper aims to characterize imaging differences between asymptomatic patients with different physical activity histories. T2 weighted MR imaging (mapping and a single image) and Dixon imaging were assessed. Dixon imaging has been used previously for imaging fat content.

It appears that the authors have used T2 weighted images rather than T2 maps. Therefore, the image results can differ between patients and results may differ if the study was performed on a different scanner. Why was a single image rather than a series of images used for this study? How can the authors ensure that the same results/findings can be seen on a different MR scanner?

A brief description of the activity histories should be provided here, even if it was previously reported. How many hours per week did individuals perform their stated activities and were they performing these activities at the time of imaging? If not, what is the time since being more active?

My major concern with using the T2 images is with the concern about being able to normalize and compare results across scanners. T2 Mapping helps to reduce the differences between scanners but if the authors want to use a single image it seems that a phantom would be needed for normalization purposes. The authors do attempt to normalize the parameter but it is with respect to the tissue itself.  

T1rho mapping has been shown to be more sensitive to early degenerative changes in the intervertebral disc than T2 imaging. Why did the authors use T2 mapping rather than T1rho?

Line 20 – “often already in the 20th” this phrasing is awkward.

While presenting the data in table format is helpful for readers that want exact values, visualization in a figure format is also greatly beneficial for comparing differences between groups. The authors should consider adding figures to visualize some of the data presented in Tables 1 and 2.

Figure 1 shows images with respect to Pfirrmann grade rather than activity level, which was the main objective of this work. The figure also suggests that Dixon imaging provides a measure of water fraction; however, the images provided do not match local measures of water content throughout the disc. Therefore, it is not likely that Dixon imaging actually represents water fraction of the tissue. This also impacts the conclusions drawn in the discussion section (e.g., Line 200).

It is not clear why correlations were performed between each imaging modality. The main objective of this work was to compare across activity levels. A statistician should be consulted to ensure that the statistical measures used match the question being presented.

My major concern with this work is the conclusions drawn with limited/no validation of the imaging techniques presented (e.g., Dixon) to image biochemical composition. Therefore, the paper is limited to T2mapping, which has been shown by numerous studies to be correlated with degeneration and activity (authors own previous work).

Author Response

Please see your comment and our reply below, as well as changes in the revised manuscript (see attached file), indicated in grey.

It appears that the authors have used T2 weighted images rather than T2 maps. Therefore, the image results can differ between patients and results may differ if the study was performed on a different scanner. Why was a single image rather than a series of images used for this study? How can the authors ensure that the same results/findings can be seen on a different MR scanner?

In this study, we used T2-maps for estimation of the relaxation time to enable consistency between measurements, scan protocols and scanners. However, the wording “T2-mapping images” was used in the manuscript instead of T2-maps. For clarity, this has been adjusted in the Material and Methods section of the revised manuscript (row 127).

A brief description of the activity histories should be provided here, even if it was previously reported. How many hours per week did individuals perform their stated activities and were they performing these activities at the time of imaging? If not, what is the time since being more active?

At your recommendation, we have included a brief description of the physical activity histories of the included individuals in the Material and Methods section (row 82-85) as follows:

“Only people with a minimum of 5 years history at their current physical activity level were included in the study: either no sport (sedentary referents), cyclists reporting a minimum of 150 km cycled per week (high-volume road cyclists), 20–40 km per week running (joggers), or 50 + km per week running (long-distance runners). See previous work [7; 8] for further details.”

My major concern with using the T2 images is with the concern about being able to normalize and compare results across scanners. T2 Mapping helps to reduce the differences between scanners but if the authors want to use a single image it seems that a phantom would be needed for normalization purposes. The authors do attempt to normalize the parameter but it is with respect to the tissue itself.  

Yes, you are correct. Singular T2 images are not normalized and do not allow comparisons across scanners. Therefore, T2-maps were used in this study. As mentioned above, this was previously unclear but has now been enhanced in the revised manuscript (Material and Methods, row 127).

T1rho mapping has been shown to be more sensitive to early degenerative changes in the intervertebral disc than T2 imaging. Why did the authors use T2 mapping rather than T1rho?

Yes, T1rho mapping has showed great promise to characterize early degenerative disc changes. Present study did not include the T1rho mapping as it still is a research technique and as such was not implemented in the MRI scanner. We encourage future studies to evaluate the feasibility of the technique for characterization of disc changes in training individuals.

Line 20 – “often already in the 20th” this phrasing is awkward.

The sentence has been corrected to “, as early as 20 years” (row 42).

While presenting the data in table format is helpful for readers that want exact values, visualization in a figure format is also greatly beneficial for comparing differences between groups. The authors should consider adding figures to visualize some of the data presented in Tables 1 and 2.

We agree that figures are greatly beneficial to display the differences between groups. To avoid redundancy of information, however, we have displayed the results only in tables. On demand, we could include one or two figures to visualize the data reported in the tables. 

Figure 1 shows images with respect to Pfirrmann grade rather than activity level, which was the main objective of this work. The figure also suggests that Dixon imaging provides a measure of water fraction; however, the images provided do not match local measures of water content throughout the disc. Therefore, it is not likely that Dixon imaging actually represents water fraction of the tissue. This also impacts the conclusions drawn in the discussion section (e.g., Line 200).

Indeed, the Dixon images did not match the T2-based images, nor globally or locally. However, the images should not match as the T2-based techniques do not only reflect the water content in the tissue but also the tissue matrix structure. In the “Discussion” section (row 206-211), this has been enhanced.

It is not clear why correlations were performed between each imaging modality. The main objective of this work was to compare across activity levels. A statistician should be consulted to ensure that the statistical measures used match the question being presented.

Yes. The main objective of this work was to compare across activity levels. However, statistical correlation analyses were also performed between imaging techniques to improve the understanding of the findings and show that both T2-based techniques display similar characteristics while Dixon imaging does not. The “Statistical analyses” section in “Material and Methods” includes a description of the correlation analyses (row 146). We have added that this analysis as well as the correlation analysis between MRI techniques and Pfirrmann grade were sub-analyses to the main comparative analysis.   

My major concern with this work is the conclusions drawn with limited/no validation of the imaging techniques presented (e.g., Dixon) to image biochemical composition. Therefore, the paper is limited to T2mapping, which has been shown by numerous studies to be correlated with degeneration and activity (authors own previous work).

T2-based imaging techniques have indeed been thoroughly validated. This work also includes a non-validated imaging technique, i.e. the Dixon technique that displays the water content pixel-by-pixel. The technique has not previously been used for disc imaging. Therefore, we have dampened the conclusions of our findings and written in the Discussion section that “Future studies are encouraged to validate this finding” (row 217).   

Reviewer 4 Report

The study titled "Characterization of intervertebral disc changes in asymptomatic individuals with distinct physical activity histories using three different quantitative MRI techniques" describes a comparative study of results between three different MRI techniques in understanding the differences in IVDs of asymptomatic individuals. The study is quite robust and gives important comparative distinction between the currently used MRI techniques, which could help in further development of these techniques for pathological evaluation. The minor comments are as below-

1) Mention both no of men and women in the study and expressing it as percentage can be avoided.

2) The formatting of Table 3 needs to be addressed.

3) The comparison of Dixon imaging with T2 and T2w should be explained further. As, the water content analysis in the paper through Dixon imaging stands alone from the rest of comparative studies.

Author Response

Please see your comment and our reply below, as well as changes in the revised manuscript (see attached file), indicated in grey.

1) Mention both no of men and women in the study and expressing it as percentage can be avoided.

The Material and Methods section has been adjusted accordingly (row 80).

2) The formatting of Table 3 needs to be addressed.

Yes, the table formatting is not similar to the submitted version of the manuscript. This will be addressed before final publication.

3) The comparison of Dixon imaging with T2 and T2w should be explained further. As, the water content analysis in the paper through Dixon imaging stands alone from the rest of comparative studies.

We agree that the comparison of Dixon imaging with T2 and T2w needs to be explained further. Therefore, we have changed the Introduction section slightly (row 68 to 71). In specific, we have changed “As such, Dixon imaging should be able to detect differences in the water content of the IVD and, thus, reflect other aspects of the tissue composition than T2-mapping and T2w imaging.” to “As such, Dixon imaging should be able to detect differences in the water content of the IVD without reflecting the structure of the IVD matrix and, thus, reflect other aspects of disc degeneration than T2-mapping and T2w imaging techniques.”. We have also clarified in that the included imaging techniques have the feasibility to detect degenerative disc behaviors (and therefore was compared in their performance) by adding the word degeneration or degenerative disc changes to row 53, 57, 65, and 70.
